# Unveiling Chemical Cues of Insect-Tree and Insect-Insect Interactions for the Eucalyptus Weevil and Its Egg Parasitoid by Multidimensional Gas Chromatographic Methods

**DOI:** 10.3390/molecules27134042

**Published:** 2022-06-23

**Authors:** Davide Mendes, Sofia Branco, Maria Rosa Paiva, Stefan Schütz, Eduardo P. Mateus, Marco Gomes da Silva

**Affiliations:** 1Associated Laboratory for Green Chemistry (LAQV) of the Network of Chemistry and Technology (REQUIMTE), Chemistry Department, NOVA School of Science and Technology, NOVA University of Lisbon, 2829-516 Caparica, Portugal; dm.mendes@campus.fct.unl.pt; 2The Forest Research Centre (CEF), School of Agriculture University of Lisbon (ISA), Tapada da Ajuda, 1349-017 Lisbon, Portugal; 3Center for Environmental and Sustainability Research (CENSE), Department of Environmental Sciences and Engineering, NOVA School of Science and Technology, NOVA University of Lisbon, 2829-516 Caparica, Portugal; mrp@fct.unl.pt (M.R.P.); epm@fct.unl.pt (E.P.M.); 4Department of Forest Zoology and Forest Conservation, Buesgen-Institute, Göttingen University, 37077 Göttingen, Germany; s.schuetz_137@gmx.de

**Keywords:** chemical ecology, GC×GC, *Gonipterus platensis*, MD-GC, pheromones

## Abstract

Multidimensional gas chromatography is, presently, an established and powerful analytical tool, due to higher resolving power than the classical 1D chromatographic approaches. Applied to multiple areas, it allows to isolate, detect and identify a larger number of compounds present in complex matrices, even in trace amounts. Research was conducted to determine which compounds, emitted by host plants of the eucalyptus weevil, *Gonipterus platensis*, might mediate host selection behavior. The identification of a pheromone blend of *G. platensis* is presented, revealing to be more attractive to weevils of both sexes, than the individual compounds. The volatile organic compounds (VOCs) were collected by headspace solid phase microextraction (HS-SPME), MonoTrap^TM^ disks, and simultaneous distillation-extraction (SDE). Combining one dimensional (1D) and two-dimensional (2D) chromatographic systems—comprehensive and heart-cut two-dimensional gas chromatography (GC×GC and H/C-MD-GC, respectively) with mass spectrometry (MS) and electroantennographic (EAD) detection, enabled the selection and identification of pertinent semiochemicals which were detected by the insect antennal olfactory system. The behavioral effect of a selected blend of compounds was assessed in a two-arm olfactometer with ten parallel walking chambers, coupled to video tracking and data analysis software. An active blend, composed by *cis* and *trans*-verbenol, verbenene, myrtenol and *trans*-pinocarveol was achieved.

## 1. Introduction

The eucalyptus weevil, *Gonipterus platensis* (Coleoptera, Curculionidae), is an invasive insect pest, causing economic damage to *Eucalyptus* production worldwide. Control usually relies on a single biological agent, the egg parasitoid, *Anaphes nitens* (Hymenoptera, Mymaridae). However, in colder regions within the range of invasion, the efficiency of the parasitoid is significantly reduced, hence, outbreaks continue to occur [1,2,3,4]. Clearly, new control alternatives are required. The use of semiochemicals in integrated pest management (IPM) programs proved to be a useful component over the past half-century, [5,6]. These compounds of natural origin, such as pheromones, are capable of attracting, or repelling, insect pests and their natural enemies. They are safe for the operator, environmentally friendly, have a short persistence in the environment and do not harm beneficial species; furthermore, they are often species-specific and can be used in very small amounts [5,6]. Different strategies can be implemented for monitoring and/or control of each pest species, such as mass trapping, mating disruption and push and pull techniques [7,8].

A comprehensive project was launched aiming at decoding *G. platensis* chemical ecology, regarding the responses to both pheromones [9] and to host volatiles [10]. Responses of *A. nitens* to *G. platensis* egg capsules, adults and their feces and to *Eucalyptus globulus*, the plant upon which *G. platensis* feeds, were also assessed [11]. *Eucalyptus* trees produce a large array of volatile compounds, mainly terpenes [12,13] showing important qualitative and quantitative variations at interspecific level, which may influence their relative attractiveness to phytophagous insects [11,14]. Indeed, phytophagous insects depend on the ability to locate a host plant, particularly if hidden among an array of non-host ones, for the fulfillment of their nutritional requirements, mate finding and location of a suitable oviposition site [15]. The first chemical stimuli that insects detect in the process of host plant selection are volatiles that act as a chemical message [16]. According to the composition of the volatile *bouquet*, a plant can be recognized as a host, eliciting insect attraction and arrestment or, conversely, be recognized as a non-host, eliciting insect repulsion and dispersal [15]. Some volatile terpenes have functions in plant communication, either at plant-plant or inter-plant level [17]. Additionally, terpenes are basic components of essential oils, and important constituents of fruit volatiles and floral scents involved in pollinator attraction [18,19,20]. Furthermore, plant emissions also contain chiral compounds, since many monoterpenes present enantiomeric forms. [21] Although not aimed in this work, it is relevant to mention that different enantiomers play an important role in biological activity [19]. Insects have olfactory receptor neurons that discriminate between enantiomers and can use them as fingerprints in host selection, or avoidance [22] while plants use them for pollinator attraction [23]. They can also be used as fingerprints in plant species chemotaxonomy studies [21].

One-dimensional gas chromatography (1D-GC), is the most widely used technique for the analysis of volatiles and semi-volatile compounds in complex biological matrices, such as the above-mentioned plant *bouquets,* or essential oils. However, in spite of being a powerful tool, sometimes the results obtained show a poor or unsatisfactory separation, due to the complexity of the sample composition. Indeed, 1D-GC is partially limited by the column’s physical and statistical separation power and consequently, the bioactive compounds may be hidden under other sample components. Sometimes one compound is very concentrated and overlaps (co-elutes) with compounds in adjacent chromatographic space, at close retention times, while sometimes bioactive compounds may even be hidden under baseline noise, if present at trace level [24]. This fact leads to misidentifications, and/or poor matches when performing MS library search, due to the “impure” mass spectra obtained. Some other problems that can frequently occur are baseline drift, spectral background, homoscedastic and heteroscedastic noise, retention time shift, non-Gaussian peaks and low S/N. Additionally, the 21st century analytic triangle (throughput, resolution and productivity) is calling and promoting the need not only for more efficient chromatographic analysis, but also for faster ones instead of slower and longer runs [25,26,27,28]. For the 1D-GC/MS systems, these new demands facing the one dimensional chromatographic reality create further problems such as misleading searches in the MS libraries and hindering accurate assignments of the components tentative identifications, whenever standards are not available [26]. Conventional MD-GC and GC×GC evolved from the traditional 1D-GC systems. The MD-GC improves the conventional 1D-GC by performing either one, or several, limited heart-cuts (H/C) in order to transfer fractions of the effluent with selected analytes, at programed sampled time ranges, from the first column (first dimension, ^1^D; obtain the full sample) to a second column (second dimension, ^2^D; obtain selected sample parts), on a single sample injection and in the same analytical run. The two columns used should have different stationary phases with complementary selectivities, in order to achieve maximum separation “orthogonality” and peak capacity [29]. Although the conventional MD-GC system uses two columns with different stationary phases (e.g., polar and non-polar), connected through an interface (e.g., Deans switch), the comprehensive exploitation of the “orthogonality” concept can only be achieved with GC×GC when the entire sample, and not just discrete parts of it, after separation in the first dimension, is subjected to separation in the second dimension. [29].

In GC×GC two capillary columns are connected in series, through a suitable interface. The sample is transferred from the first (^1^D) to the second (^2^D) column, frequently without loss of mass. The interface comprises a modulator, which enables the sequential liberation of the solute from the first column onto the second column, preserving and further fractionating the separation obtained on the first one. This allows for all sample components to be subjected to two different separation processes, resulting in a drastic increase in peak capacity and separation efficiency of the chromatographic system, as well as in the detectability of analytes. Since the introduction of GC×GC in 1991 by Liu and Phillips [30], the field of gas chromatography has seen an explosion in innovation [31], thus providing analytical laboratories with powerful techniques to improve data acquisition and subsequently data treatment tools, such as machine learning and artificial intelligence [32,33,34].

The study of volatile organic compounds (VOCs), used by insects as cues for host location, or in intraspecific communication, usually follows a sequence of steps: (1) Collection of the insect- and/or host plant-emissions. When pheromone sampling is targeted, previous knowledge of the insect behavior is needed, so that sampling can be timed with the putative emission of the compounds, or blends of interest; (2) Determination of compounds or blends emitted, which are detected by the olfactory system of the insect; this is usually achieved using a method known as electroantennography (EAG). EAG is based on the finding of Schneider (1957), in that when insect antennae are exposed to adequate stimuli, changes in their slow electrical potential occur [35]. These electrical potential alterations can then be measured by placing recording electrodes in the hemolymph, at the tip and the base of an antenna [35,36,37]. Coupling an electroantennographic detector to a gas-chromatography/mass spectrometry GC/MS system, (GC/MS-EAD setup), allows for the simultaneous determination of which VOCs have sensory activity in the antenna and their chemical identification, all in one run [38,39]. The final step is testing which of the compounds, or blends, detected by the insect have behavioral activity [40].

This work describes how combining 1D and 2D systems (GC×GC and MD-GC) with mass spectrometric (MS) and electroantennographic (EAD) detectors, enabled the detection and identification of semiochemicals that elicit behavioral and physiological responses in tri-trophic interactions among *G. platensis*, its host plant and its egg parasitoid. Our research hypothesis is that not all active compounds can be well resolved under the standard 1D chromatographic conditions associated to the GC/MS-EAD system.

We further hypothesize that the aggregation pheromone of *G. platensis* consists of a combination of compounds, and not just of individual compounds as previously reported [8]. Here, we present the identification of a pheromone blend of *G. platensis*, putatively more attractive to weevils of both sexes than the individual compounds previously identified.

## 2. Results

### 2.1. GC/MS-EAD: Detection and Identification of Bioactive Compounds

The GC/MS-EAD analysis conducted with *G. platensis* led to the detection and putative identification of 61 compounds perceived by the olfactory system of the weevils (Table 1). Ten of these compounds are exclusively produced by *G. platensis* [9], whereas the remaining 51 are emitted by either *E. globulus* [10], or by both *E. globulus* and *G. platensis* (Table 1). Three compounds were shown to be emitted only by males (verbenene, *cis*-verbenol and *trans*-verbenol) [9]. Both the compound identities and electrophysiological activities were confirmed for 40 compounds. The parasitoid displayed antennal response to 45 compounds: ten emitted exclusively by *G. platensis* adults, 28 emitted by *E. globulus*, or by both *E. globulus* and *G. platensis* adults, feces and egg capsules, and five compounds that were only detected in *G. platensis* egg capsules and feces: 2-heptenal, 6,7-dimethyl-1,2,3,5,8,8a-hexahydro-naphthalene, 2,9-dihydroxy-1,8-cineole and 2,6-dihydroxycineol [11]. The full characterization of these volatiles can be consulted elsewhere [9,10,11].

### 2.2. Circadian Variation of G. platensis Emitted Compounds

The analytical data uncovered synchronized fluctuations in the emission of the *G. platensis* male-specific compounds with strong correlations among verbenene, *cis*-verbenol and *trans*-verbenol (*r* values ≥ 0.90, Pearson product-moment correlation coefficients) (Appendix A). Additionally, synchronized fluctuations between these compounds and *trans*-pinocarveol and myrtenol were found. Thus, data support the hypothesis that hourly variations detected in the emissions follow a common pattern. Myrtenol, just like *trans*-pinocarveol, also elicited electroantennographic responses from *G. platensis*, as confirmed by standard co-injection.

### 2.3. Behavioral Bioassays

Behavioral responses of *G. platensis* to individual compounds, identified as bioactive in GC/MS-EAD analysis, were tested in previous studies [9,10]. Putative components of an aggregation pheromone were identified: both virgin females and virgin males were attracted to the male-specific compound *cis*-verbenol, while virgin females were further attracted to the *trans* isomer [9]. Five compounds, emitted by males, showed strong correlations (*r* ≥ 0.90) in the circadian variation assay, and were thus used to create reconstructed synthetic compound blends, according to their relative composition, determined by GC×GC-FID. For an extract of *G. platensis* virgin males, which proved attractive to virgin females in olfactometer bioassays [9], the relative percentage of the emitted compounds with correlation coefficients ≥ 0.90 were: verbenene 3.32%, *trans*-pinocarveol 5.76%, *cis*-verbenol 13.83%, *trans*-verbenol 53.64% and myrtenol 23.45%. This information was used to produce blends containing these five compounds in approximately the same ratio as in the attractive male-produced extract, Table 2.

The bioassays started with *cis* + *trans*-verbenol at a total concentration of 10^−5^ (*w*/*v*), followed by the addition of one component at a time, in sequence: verbenene, then myrtenol and finally trans-pinocarveol. At all times, the percentages of the different compounds in the mixture corresponded to those determined for the male emissions. In total, four combinations were tested. The time spent by virgin adult weevils in the treatment arm, was always higher than the time spent on the control arm in all tested combinations. However, statistically significant differences were only observed for the final, five component mixture (*n* = 40, *p* = 0.016 for virgin females; *n* = 60, *p* = 0.033 for virgin males).
molecules-27-04042-t002_Table 2Table 2Olfactometer responses of *G. platensis* to compound blends, expressed as the mean time ± SD spent in the treatment arm (+) vs. control (−) in seconds (Mann Whitney U tests, *p* < 0.05). Bold indicates that significant differences were observed between the time spent in the treatment and control arms. * Denote *p* < 0.05.Time Spent in Arm (s)Virgin FemalesMated FemalesVirgin MalesMated Males
+−+−+−+−Compound Blend







*cis* + *trans*-Verbenol 601 ± 249482 ± 267448 ± 323657 ± 315566 ± 301452 ± 303478 ± 394572 ± 419*cis* + *trans*-Verbenol + verbenene631 ± 371455 ± 356489 ± 347573 ± 343635 ± 358442 ± 363

*cis* + *trans*-Verbenol + verbenene + myrtenol595 ± 341488 ± 336500 ± 255619 ± 254603 ± 390457 ± 376485 ± 331543 ± 362*cis* + *trans*-Verbenol + verbenene + myrtenol + *trans*-pinocarveol**672 ± 328** *
**435 ± 310**550 ± 398511 ± 390**653 ± 371** *
**446 ± 358**520 ± 281558 ± 288

To determine if there were differences in the mean time spent in the treatment arm, between the final compound blend and components of the mixture that were individually attractive [9], Mann Whitney U tests were performed. The attractiveness of the full blend was therefore compared to the attractiveness of *cis*-verbenol for virgin males and virgin females and compared to the attractiveness of *trans*-verbenol for virgin females: significant differences were found for virgin males (*n* = 120, *p* = 0.027), the full blend being more attractive than *cis*-verbenol alone. The strong temporal correlations found among the emissions of five compounds, support the hypothesis that they are simultaneously emitted and may constitute potential components of the male pheromone.

## 3. Discussion

### 3.1. The Co-Elution Problem

GC/MS-EAD was performed in a 1D separation on a polar Wax type column. Thus, the absence of co-elutions that could hide true bioactive compounds (which, given their specificity, might be present in trace amounts), cannot be guaranteed. In all samples analyzed, co-elutions were detected under the operational conditions of the GC/MS-EAD setup by visual observation of peak shoulders and absence of mass spectra uniformity across peaks. In addition to complete co-elution, some peaks showed a resolution lower than 1, implying that base line separation was not achieved [41]. This may be due to the high concentration of some of the compounds hidden under adjacent peaks. Such was the case for the observed co-elution between *α*-terpinene, limonene and 1,8-cineole, in the SDE sample of *E. globulus*. It was due to the high concentration of 1,8-cineole and to the co-elution between 2-*α*-hydroxy-1,8-cineole and 9-hydroxy-1,8-cineole in weevil extracts, resulting from the high concentration of the 2-*α*-hydroxy isomer. In order to clarify the identity of the co-eluting compounds present in the chromatographic space of interest, MD-GC/MS and GC×GC-FID were used to re-analyze the extracts. The observed increase in peak capacity proved that both analytical techniques may decisively contribute to clarify the composition of the samples analyzed, as well as to a more robust elucidation of the peaks identities achieved. By using GC×GC-FID and MD-GC with a Dean switch—heart-cutting device—co-elutions were resolved and the compounds present in chromatographic space of the GC/MS-EAD analysis where antennal response was observed, could be identified.

### 3.2. GC×GC-FID

GC×GC is based on the same principles as H/C, yet applied to the full extent of the chromatogram and performed through a fast and continuous heart-cutting-modulation—with a sampling period smaller than the width of a ^1^D peak, for non-discriminating analysis [42,43]. The modulator is a crucial component for a successful GC×GC analysis. The three primary functions of the modulator are: (i) to accumulate or trap; (ii) to refocus; (iii) to release narrow adjacent zones of ^1^D effluent rapidly into 2D [42,43]. Since the flow in the ^1^D is constant, the modulation period (P_M_) also needs to be constant, rapid, and precise to preserve the ^1^D separation throughout the instrument run [42,43]. On the ^2^D column, the collected sample from the modulator needs to be eluted before the following collected sample is released from the modulator to minimize the occurrence of wrap-around [42,43]. Due to the fast elution in the ^2^D, the detector must have a high-speed acquisition rate (at least 50 Hz), especially due to the resulting narrow peaks—peaks width of 50–300 ms at the base [44]. When the GC×GC is applied, an increase in the peak capacity (n_c_) occurs equal to the product of the peak capacity of both ^1^D and ^2^D (n_c1_ × n_c2_) [44]. The ^1^D has a longer length and a higher n_c_ value, in contrast to ^2^D, which needs to have a shorter length to permit a very fast elution, so that the two-dimensional run is completed within practically the same time as the ^1^D run [44]. Hence, in order to further unravel the composition of the samples and to detect chromatographic co-elutions that may still hide the bio-active compounds from the GC/MS-EAD analysis, GC×GC-FID qualitative and semi-quantitative experiments were also performed on a polar × non-polar column set, where the first polar column is equivalent to the WAX column used in the GC/MS-EAD experiments. Figure 1 shows a representative GC×GC-FID chromatogram (2D color plot) of a sample prepared after HS-MonoTrap™, for virgin males of *G. platensis*.

Plot comparisons were made between: (i) *G. platensis* virgin males; (ii) *E. globulus* leaves and shoots; (iii) virgin males placed together with *E. globulus* leaves and shoots. Co-elution of *trans*-pinocarveol (compound 31) and *cis*-verbenol (compound 32) was confirmed and resolved. However, although the compounds present in the extracts were separated by GC×GC-FID, structural information could not be obtained since an FID detector was used. For this reason, it was necessary to submit the samples to mass spectrometry under operating conditions that mimic the GC×GC run.

### 3.3. MD-GC with Dean Switch—Heart-Cutting

MD chromatography is characterized by an interface and controls that allow the coupling of analytical columns with different chemical characteristics and dimensions [42]. Hence, the MD-GC analytical process selects a (bounded) zone or region of the compounds eluted at the end of a GC column and subsequently subject it to another GC separation. In 1968 D.R. Deans proposed Deans switch based on pressure switching, which brought about several advantages, such as absence of temperature limitations, formation of artifacts, memory effects or direct contact between parts of the mechanical valve with the sample compounds [45]. In multidimensional heart-cut chromatography usually a column is used as the first dimension (^1^D) and connected through the heart-cut valve, to at least two other columns with stationary phases different from the first one. This topic was comprehensively revisited in the last 10 years [46,47]. After eluting from the ^1^D column, two possible routes can be used by the eluting compounds: directly to an FID detector, through a deactivated capillary column without further separation or to a second dimension column (^2^D). The splitting flow is achieved by the Deans switch device which coupled the ^1^D column to the deactivated capillary column and to the ^2^D column, containing a stationary phase of different polarity than the ^1^D column. The deactivated column preserves the separation, achieved in the ^1^D column, until the FID detector is reached. A chromatogram is then obtained, based on the separation of the ^1^D column, that simultaneously ensures the control of the working pressures, thus avoiding backflush. The deactivated column preserves the separation, achieved in the ^1^D column, until the FID detector is reached. A chromatogram is then obtained, based on the separation of the ^1^D column, that simultaneously ensures the control of the working pressures, thus avoiding backflush. Since the heart-cut technique targets only a fraction of the sample to the ^2^D column, the separation power of the ^1^D column increases. The Deans switch valve renders possible to perform cuts, by directing the flow to the valve, as sketched on Figure 2. In this work, a MD-GC with H/C analyses was performed using a configuration that mimics the column set of GC×GC-FID. The first dimension comprises a WAX column, connected to an FID through a deactivated column, and the second dimension a 5% phenyl-methyl-polydimethylsiloxane column, connected to a mass selective detector (MSD), aimed at obtaining structural information (Figure 2). In order to optimize the operational conditions that enable accurate H/C transfers, operation at constant flow.

Constant flow (CF) and constant pressure (CP) were assayed. For this purpose, a mixture of hydrocarbon standards (C8–C20), a Grob mixture and a mixture of three compounds (1-octen-3-ol, guaiacol and indole), allowing a retention time spread over the chromatographic run, were used. In both cases, CF and CP good linearity was obtained throughout the run (Appendix A, respectively).

Running a hydrocarbon mixture performing H/C, an accurate time interval for the cut was established, after the estimation of the time that each compound takes to travel the entire length of the deactivated column, assuming, for calculation purposes, that no chromatographic separation occurs. Under CF conditions, the “dead volume” time interval is constant, due to the inherent fact that the method has a constant flow and, consequently, the linear velocity of the carrier gas is also constant. However, the experimental “travel time” for the compounds from the DS to the detector in the ^1^D decreased along the run time. This was probably due to a higher volatilization of the compounds and lower interaction with the surface of the deactivated column as the temperature increased across the chromatographic run.

Under CP conditions, there was a gradual increase in the time interval, due to the fact that both the flow and the linear velocity of the carrier gas gradually decreased with the temperature rise, as seen on Appendix A. The calculations for the H/C time period also took in consideration the peak broadening (in minutes), in order to guarantee that the transferences from ^1^D to ^2^D columns were quantitatively accurate. Indeed, based on the analysis of the variation of the “dead volume” time throughout the run, for both CP and CF flow modes (Appendix A) linear equations were obtained and indicated in the respective Figures.

These equations were used to determine the period of time of each cut for the target compounds, to which 0.05 min and 0.10 min were added, before and after the peak, respectively, considering the band width of each target compound. This procedure allowed to anticipate any readjustment during all the chromatographic run, ensuring a quantitative transference from the ^1^D to the ^2^D column. Since the control system of the GC had only one decimal number on the flow display, the precision of the electronic flow control (EFC) was low, as can be observed from the irregularity of Appendix A plot. Thus, keeping the pressure constant simplifies the cutting procedure, yet requires optimization.

Indeed, in CP mode, it is necessary to carry out constant adjustments based on the chromatographic run time, but on the other hand, it is possible to operate at lower pressures, compared to the CF operation. Considering the operational data obtained, CP operation was chosen for further application. Additionally, a Grob mixture and a cork sample extract spiked with a mixture of three components (1-octen-3-ol, guaiacol and indole) were used in order to test and verify the practical effectiveness of the H/C procedure [48].

The use of MD-GC enhanced the 1D-GC chromatographic capabilities. The addition of a ^2^D column sequentially to the ^1^D column via a Deans switch system, increased the total peak capacity of the system, since the peak capacity (n_c_) of the ^2^D column (n_c2_) is added to the n_c_ of the ^1^D column (n_c1_). With this technique, it was possible e.g., to verify in an EAG active space, where it could not be unequivocally determined if the compound present was α-terpineol (compound 35) or terpinyl acetate (compound 36), that both compounds were present in the extract. The fact that both compounds simultaneously co-elute and have very similar mass spectra makes it difficult to determine which compound is present. By means of MD-GC/MS it was possible to chromatographically resolve this co-elution and verify, through their „clean” mass spectrum, that both *α*-terpineol and terpinyl acetate were present in the sample Additionally, the multidimensional data also show that the two compounds were co-eluting with ledene (compound 37) in 1D-GC (Figure 3). This methodology was repeated for all similar cases and the identification assignments used to build Table 1, since standard compounds were used to confirm compound identities and the identifications obtained by MS spectral libraries.

## 4. Materials and Methods

### 4.1. Reagents and Materials

(*Z*)-3-Hexen-1-yl acetate (98%), (+)-*α*-pinene (>99%), (−)-*α*-pinene (98%), camphene (95%), (+)-*β*-pinene (98%), (−)-*β*-pinene (98%), *β*-myrcene (95%), (*R*)-(+)-limonene (97%), (*S*)-(−)limonene (96%), 1,8-cineole (99%), *γ*-terpinene (97%), *p*-cymene (97%), *α*-terpinolene (≥90%), linalool oxide (≥97%), sabinene hydrate (≥97%), (−)-*α*-gurjunene (≥97%), (+)-calarene (≥99%), *β*-caryophyllene (98,5%), (+)-aromadendrene (≥97%), (−)-*trans*-pinocarveol (≥96%), (+)-fenchol (analytical standard), (−)-*allo*-aromadendrene (≥98%), *α*-terpineol (90%), *α*-terpinenyl acetate (≥95%), (+)-ledene (≥95%), (*1R*)-(−)-myrtenol (95%), (−)-carveol (97%), 2-phenylethanol (≥99%), phenylethyl isovalerate (≥98%), (−)-*epi*-globulol (≥95%), (−)-globulol (≥98.5%), (*S*)-(−)-perillyl alcohol (≥95%), (*S*)-*cis*-verbenol (95%), (1*S*)-(−)-verbenone (94%) and paraffin oil (for IR spectroscopy) were purchased from Sigma-Aldrich (Steinheim, Germany). Regarding the synthesized compounds: (+)-trans-verbenol was prepared from (+)-*α*-pinene [49]; verbenene synthesized from verbenone [50,51]; 2-*α*-hydroxy-1,8-cineole and 2-oxo-1,8-cineole were prepared according to the procedures previously described [52]; 9-Hydroxy-1,8-cineole was synthesized from *R*-(+)-limonene [53]. Hexane (99%) was supplied by JT Baker (Deventer, The Netherlands), dichloromethane (Suprasolv grade) and anhydrous sodium sulphate by Merck (Darmstadt, Germany) and (*Z*)-3-hexen-1-ol (98%), ocimene (mixture of isomers, ≥90%), (−)-*α*-copaene (>90), *α*-phellandrene (≥85), geraniol (≥98%) and the hydrocarbon mixture (C8–C20) by Fluka (Neu-Ulm, Germany).

### 4.2. Insect Rearing and Plant Sampling

*Gonipterus platensis* was reared as previously described [8,9,10]. *Anaphes nitens* was reared as described elsewhere [10]. Samples of *E. globulus* foliage were collected in plots located in the councils of Figueiró-dos-Vinhos, Ferreira do Zêzere, Tomar, Loures, Arouca and Caparica, in Portugal, and extracted within 24 h after collection.

### 4.3. Collection of VOCs

#### 4.3.1. Headspace Monolithic Material Sorption Extraction (HS-MMSE)-MonoTrap™ Disks

MonoTrap^TM^ disks (MT; MonoTrap-DCC18-Activated Carbon, ODS, End-Capped; GL Sciences, Tokyo, Japan) were used for MMSE headspace extractions. Extracts from live *G. platensis* were thus obtained which were used for characterization of the volatile fraction and for electrophysiological and behavioral bioassays. Insects two to four months old were sampled following a described procedure [8,9,10]: the sampling material was placed inside 250 mL glass jars, the MT disk suspended at a fixed position in the center of the jar lid and left extracting for 48 h. After sampling, 200 µL of dichloromethane was used as extraction solvent. All extracts were stored at −20 °C until use. The following combinations were sampled: (i) 10 virgin females; (ii) 10 virgin males; (iii) 10 mated females; (iv) 10 mated males, (v) *E. globulus* leaves and shoots; (vi) 10 virgin females with *E. globulus* leaves and shoots; (vii) virgin males with *E. globulus* leaves and shoots.

#### 4.3.2. Simultaneous Distillation–Extraction (SDE)

SDE was conducted with 100 g of fresh *E. globulus* leaves and shoots as described elsewhere [8,9,10]. The foliage was placed in a round-bottom flask (0.5 L) with twice distilled water (Milli-Q RG, Millipore, Molsheim, France; 350 mL). The flask was connected to a Veith and Kiwus exhaustive steam-distillation and solvent-extraction apparatus, and extraction was performed for 2 h with 10 mL hexane. The final extract was dried with anhydrous sodium sulfate and stored at −20 °C until use.

#### 4.3.3. Extraction with Dichloromethane

Solvent extraction with dichloromethane (DCM) was performed for *G. platensis* egg capsules, mated female feces, virgin female feces and male feces (mated + virgin) [10]. The samples were prepared by adding 15 mL of DCM to 0.5 g of fresh feces or egg capsules in 22 mL vials (Supelco, Bellefonte, PA, USA). The vials were then wrapped in aluminum foil, stored at 4 °C for 24 h and after this period, the solutions were filtered using Acrodisc PTFE Syringe Filters (0.2 µm pore size; 25 mm; Pall Gelman, Ann Arbor, MI, USA).

#### 4.3.4. Headspace Solid-Phase Microextraction (HS-SPME)

HS-SPME sampling of *E. globulus* was conducted as previously described [9]. Sampling of *G. platensis* males and *G. platensis* females with and without *E. globulus* shoots as food source [8] and HS-SPME sampling of egg capsules [10] was conducted. Briefly, polydimethylsiloxane (PDMS) and 50/30 μm divinylbenzene/carboxen/polydimethylsiloxane (DVB/CAR/PDMS; #57329-U) coated fibers were used. Before the analysis, the fiber was conditioned according to the manufacturer’s standard procedures. Extractions were performed at room temperature (23 °C) for 45 min. For egg capsules, extractions lasting 17 h were also performed. Analytes were thermally desorbed from the fiber at 250 °C in the chromatograph injection port, for 3 min.

In order to ascertain if the emission of specific volatiles by *G. platensis* would follow a circadian rhythm, samples were collected at hourly intervals for consecutive periods of 24 h, using groups of 10 wild-caught males each. The VOCs emitted were extracted using a CTC Analytics CombiPal automatic sampler with a SPME holder equipped with a divinylbenzene/Carboxen/Polydimethylsiloxane fiber. Each group of weevils was placed with fresh *E. globulus* shoots, inside 22 mL vials. The PTFE septa of the vials were modified in order to accommodate a custom-built conical wire mesh. The photoperiod lasted from 06:00–21:00 during sampling time. Three replicates were made, with three groups of ten insects each, over three consecutive days. The headspace extraction was performed at room temperature (23 °C) for 45 min and the trapped compounds desorbed at 250 °C in the chromatograph injection port, for 3 min.

### 4.4. Gas Chromatography/Mass Spectrometry (GC/MS)

For analysis of *G. platensis* and *E. globulus* samples, volatiles were separated on both non-polar (DB-5) and polar (DB-Wax) columns on three GC/MS systems:

*System 1:* consisted of a Hewlett Packard HP 5890 Series II Gas Chromatograph connected to an HP 5972 Series Mass Selective Detector (Agilent Technologies, Palo Alto, CA, USA). Analysis was performed on a DB-5 (5% phenyl–95% methyl-polysiloxane), with 30 m × 0.32 mm i.d. × 1.00 μm film thickness (d_f_) (J&W Scientific, Folsom, CA, USA). The oven program started at 40 °C for 1 min and increased to 270 °C at a rate of 6 °C·min^−1^, where it held for 10 min. The carrier gas was helium at a flow rate of 1 mL·min^−1^ in constant flow mode. MS operated at an electron ionization of 70 eV, data acquisition by full scan at a range of 40–300 Da. The source and transfer line were set at 250 °C. Instrument control and data acquisition were conducted with HP ChemStation Software (Version E.02.01.1177).

*System 2:* consisted of a Bruker 456 GC connected to a Bruker Scion TQ Mass Selective Detector (Bruker, Billerica, MA, USA). The analytical column used was a Zebron DB-Wax plus (Polyethylene glycol stationary phase) with 30 m × 0.25 mm i.d. × 0.25 μm d_f_ (Phenomenex, Torrance, CA, USA). The oven temperature program started at 40 °C for 2 min and increased to 210 °C at 4 °C·min^−1^, where it held for 9.5 min. Carrier gas was helium at a flow rate of 1.2 mL·min^−1^ in constant flow mode. Manual SPME injections were performed, in splitless mode for 3 min, in a Bruker Programmable Temperature Vaporizing (PTV) injector (starting at 40 °C with ballistic increase to 250 °C and held the entire run). When solutions were injected, 1 µL of sample was used, under the same PTV injection conditions described above. MS operated at electron ionization of 70 eV, data acquisition by full scan at a range of 40–300 Da. Source and transfer line were set at 250 °C. Instrument control and data acquisition were conducted with MSWS 8.2 Bruker and analyzed with Bruker MS Data Review 8.0.

*System 3:* in this system, a Bruker 456 GC with a Bruker Scion TQ Mass Selective Detector (Bruker, Billerica, MA, USA) were used. The analytical column was a ZB-5 (5% phenyl–95% methyl-polydimethylsiloxane), with 30 m × 0.25 mm i.d. × 0.25 μm d_f_ (Phenomenex, Torrance, CA, USA). The oven temperature program started at 40 °C, increased to 180 °C at a rate of 4 °C·min^−1^ and increased from 180 °C to 250 °C·min^−1^ at a rate of 20 °C·min^−1^. Final temperature was held for 2.5 min. This system was used to obtain a preliminary assessment of eventual daily variations in male volatile emissions and therefore the oven temperature ramp was adjusted to allow for clear deconvolution of the targeted compounds. The carrier gas was helium at a flow rate of 1.7 mL·min^−1^ in constant flow mode. HS-SPME automatic injections were performed in splitless mode at 250 °C, every hour for three consecutive days, in a Bruker PTV Injector, using the conditions described above by system 2. All the remaining GC/MS parameters and data analysis were as described for system 2.

All analytes, in all systems, were tentatively identified using retention index criteria (Linear Retention Index—LRI), mass spectra/LRI libraries or related publications [52,53,54,55,56,57,58,59]. Linear retention indices (LRI) were estimated according to van den Dool and Kratz [60] by injecting 1 µL of a hydrocarbon mixture (C8–C20; Fluka, Neu-Ulm, Germany).

### 4.5. MD-GC/MS Analysis

The analysis was performed on an MD-GC/MS system consisting of a Bruker GC 456 connected to a Bruker mass selective detector Scion TQ (Bruker, Billerica, MA, USA). An automatic sample injector was used: CTC Analysis auto-sampler CombiPAL (Agilent, Santa Clara, CA, USA). Data were acquired with MSWS 8.2 Bruker and analyzed with Bruker MS Data Review 8.0. The chromatographic separation was achieved on a DB-WAX capillary column (30 m × 0.25 mm i.d., 0.25 μm d_f_) supplied by Agilent (Santa Clara, CA, USA) as 1st dimension column (^1^D), connected by means of a Dean switch with a DB-5 capillary column (30 m × 0.25 mm i.d., 0.25 μm d_f_) supplied by Agilent (Santa Clara, CA, USA) as 2nd dimension column (^2^D) which was directly connected to the mass spectrometer (MS) and a deactivated silica column (30 m × 0.25 mm i.d.) supplied by Bruker (Billerica, MA, USA) linking the ^1^D column outlet to the FID. Appendix A shows the column arrangement inside the oven. The chromatographic conditions were: initial temperature of 40 °C for 1 min, raised at 4 °C·min^−1^ up to 240 °C, and held for 9 min. Helium was used as carrier gas at a constant pressure of 35 psi (EFC 21) and 23 psi (EFC 24). The MS transfer line and source temperatures were set at 240 °C and 220 °C, respectively. The FID was set at 250 °C.

### 4.6. GC×GC-FID Analysis

A LECO GC×GC system (LECO Corp.) with a quad jet LN_2_ (liquid nitrogen) modulator and an Agilent 7890 (Agilent Technologies) GC equipped with a FID was used, following the methodology previously described [10]. The data collection rate was set to 50 Hz, and the data were processed using LECO Corp. ChromaTOF^TM^ software. The 1D column was a polar DB-WAX (30 m × 0.25 mm i.d. × 0.25 μm d_f_; Zebron-Phenomenex, Torrance, CA, USA), and the 2D column was a non-polar DB-5 (1.75 m × 0.15 mm i.d. × 0.15 μm d_f_; Bruker, Billerica, MA, USA). The modulation period was set at 4.0 s. The main oven (which housed the 1D column) was programmed to start at 50 °C, hold for 0.5 min, and raise to 220 °C at 5 °C·min^−1^. The maximum temperature was held for 10 min. The second oven temperature program was similar to the main oven, but held 5 °C above the first one. Hydrogen was used as carrier gas, supplied at a 1.0 mL·min^−1^ at constant flow. All injections (1 μL) were performed in splitless mode for 1 min, at a constant injector temperature of 250 °C. The FID system operated at 250 °C.

### 4.7. Gas Chromatography–Mass Spectrometry/Electroantennographic Detection (GC-MS/EAD) Analysis

GC-MS/EAD analysis was conducted as previously described [9,10,11]. An Agilent system (Santa Clara, CA, USA) was used, consisting of a type 6890N GC simultaneously connected to a type 5973N quadrupole mass spectrometer containing a polar INNOWAX column (30 m × 0.25 mm × 0.25 µm d_f_) and with an antenna holder [61] via an “olfactory detector port” (ODP-2, Gerstel, Linthicum, MD, USA). Data acquisition took place with the MS ChemStation software (Agilent, Santa Clara, CA, USA). The effluent from the column was split into two pieces using a Graphpack 3D/2flow splitter. One of the capillaries lead to the MS, the other one to an “olfactory detector port” (ODP-2, Gerstel, Linthicum, MD, USA). The GC run program began at 50 °C, held for 1.5 min, ramped to a final temperature of 250 °C at 7.5 °C·min^−1^ and held for 5 min. Helium was used as carrier gas at a flow rate of 1 mL·min^−1^ in constant flow mode Injection of 1 µL of each sample was performed using a 7163 autosampler.

Freshly excised antennae of *G. platensis* and *A. nitens* were individually placed in an antenna holder. Both ends of the antenna were in touch with an electrolyte solution, which in turn was in contact with a pair of Ag/AgCl electrodes. The electronic signals obtained from the antennas were transferred to a setup that amplified these signals by 100× and was recorded by Agilent GC Chemstation software.

For *G. platensis*, antennas were tested using *E. globulus* SDE extract, HS-Monotrap™ extract of *G. platensis* virgin males, and HS-Monotrap™ extract produced by *G. platensis* males in contact with *E. globulus* leaves and shoots [9,10]. For *A. nitens* GC-MS/EAD runs were conducted with all the samples described in Section 4.3, with the exception of HS-Monotrap™ extracts of *G. platensis* in contact with *E. globulus* leaves and shoots [11].

### 4.8. Behavioral Bioassays

To test the behavioral responses of *G. platensis* to the compound blends a custom built two-arm olfactometer (Marburg, Germany) with ten individual walking chambers was used as described in Branco et al. [9,10]. Insects were inserted in the middle of the walking chambers and their position recorded for 20 min with the help of a Logitech-Webcam QuickCam Pro 9000 (Logitech, Europe, S.A., Lausanne, Switzerland) connected to Ethovision^®^ XT 8.0 software (Noldus Information Technology Inc., Leesburg, VA, USA). A minimum of 40 insects of each sex and physiological condition (40 virgin males, 40 virgin females, 40 mated males and 40 mated females) were used.

A treatment was considered attractive when the weevils spent significantly more time (according to results of the statistical tests applied, described in Section 4.9) in the arm containing the blend than in the control arm. Between trials, the olfactometer was washed with methanol, allowed to dry at room conditions, and the position of blend and control inverted to avoid positioning bias.

### 4.9. Statistical Analysis

Data analysis was performed with IBM SPSS^®^ statistics 20 software (IBM Analytics, Armonk, NY, USA). For analysis of the circadian variation of insect emitted compounds, a Pearson product-moment correlation coefficient, *r*, was estimated for the chromatographic data pertaining to the three days sampling period. For the olfactometer essays significant differences (*p* values ≤ 0.05) between the insect’s responses to stimuli and to control volatiles were determined using either paired samples *t*-test, or Wilcoxon Signed Ranks Test, whenever assumptions required by parametric statistics were not fulfilled. To determine if there were differences in the mean time spent in the treatment arm, between attractive compounds tested individually [9] and the final compound blend, a Mann Whitney U test was used.

## 5. Conclusions

Multidimensional gas chromatographic techniques, such as heart-cut MD-GC and GC×GC were used to unveil complex biological matrices yielding important results for the trans-disciplinary field of chemical ecology. The association between mass spectrometry data and electroantennographic detection support the hypothesis that *G. platensis* males emit a sexual/aggregation pheromone comprising five compounds: verbenene, *cis* and *trans*-verbenol, *trans*-pinocarveol and myrtenol. *Cis*-verbenol is likely the main component of this blend as the compound alone is able to attract both virgin males and virgin females. Nevertheless, the blend proved to be more attractive to virgin males than only *cis*-verbenol, and opens new possibilities for the development of monitoring and control strategies for this important pest of *Eucalyptus* plantations. Heart-cut MD-GC through discrete pertinent heart-cuts enable, in a simple and automatable way, to extract relevant target information, thus confirming its up-to date usefulness to characterize complex samples.

## Figures and Tables

**Figure 1 molecules-27-04042-f001:**
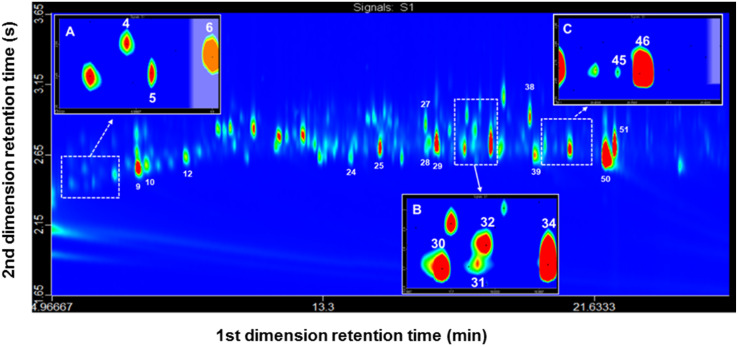
GC×GC-FID chromatogram (2D color plot) of an extract obtained by HS-MonoTrap™ of *G. platensis* virgin males. (**A**) Close up at retention time 5.2–7 min; (**B**) Close up at retention time 17.35–18.35 min; (**C**) Close up at retention time 20.1–21.6 min. Analytical conditions are described in Section 4.6. Peaks are numbered according to Table 1.

**Figure 2 molecules-27-04042-f002:**
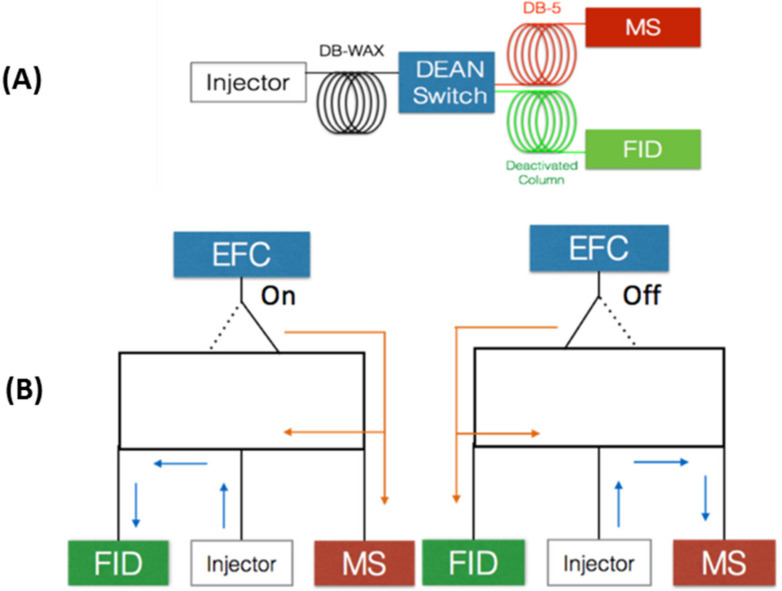
Column configuration for MD-GC system (**A**) and Dean switch schematics (**B**). Deans switch valve working diagram. EFC denotes for electronic flow control. Arrows denote the flow direction when EFC is switched “on”, flow directed to FID or “off”, flow directed to MS.

**Figure 3 molecules-27-04042-f003:**
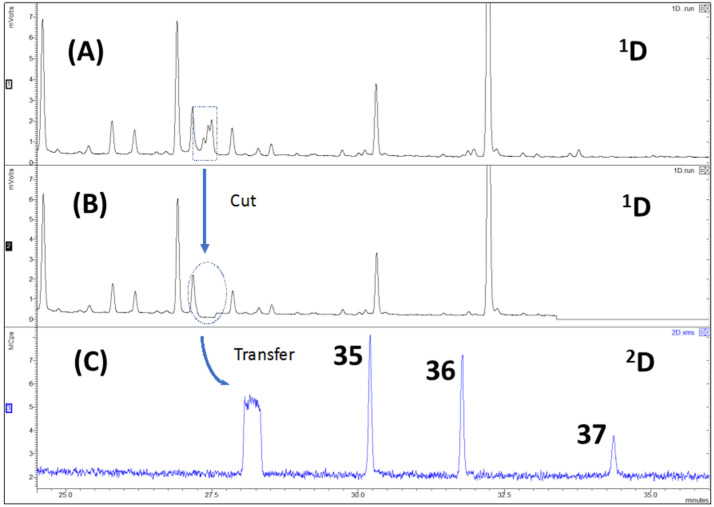
Example of chromatograms obtained by MD-GC of a HS-MonoTrap™ sample of *E. globulus*: (**A**) FID chromatogram; (**B**) FID heart-cut chromatogram; (**C**) TIC of the performed cut. Peaks are numbered according to Table 1. Analytical conditions are described in Section 4.5.

**Table 1 molecules-27-04042-t001:** Putative ID of the compounds inducing an EAG response by G. platensis, their retention time (RT) in an INNOWAX column (30 m × 0.25 mm i.d. × 0.25 µm d_f_), and presence, or absence, in the extracts used in GC/MS-EAD analysis. For each sample, co-elutions in the GC/MS-EAD operational conditions are signaled by *. Confirmation of compound identity and antennal activity by standard injection is indicated with a (+) sign in the “Standard” column. The (-) sign indicates that a Standard was not available to confirm compound identification and antennal activity. The presence or absence of each compound in the extracts used in GC/MS-EAD recordings was confirmed by GC×GC-FID and MD-GC/MS.

Compound No.	Putative ID	RT Innowax	*E. globulus*	*E. globulus* + ♂	♂	Standard
1	*α*-Pinene	6.11	+	+	+	+
2	Camphene	6.33	+	+	+	+
3	Methyl 2-methylpentanoate	6.59	+	-	-	-
4	*β*-Pinene	6.84	+	+	+	+
5	Verbenene	7.33	-	+	+	+
6	*β*-Myrcene	7.78	+ *	+	+	+
7	*α*-Phellandrene	7.88	+ *	+	+	+
8	*α*-Terpinene	8.23	+ *	+	+	-
9	Limonene	8.63	+ *	+	+	+
10	1,8-Cineole	8.95	+ *	+	+	+
11	*γ*-Terpinene	9.49	+	+	+	+
12	*p*-Cymene	9.91	+	+	+	+
13	*α*-Terpinolene	10.11	+	+	+	+
14	*neo-allo*-Ocimene	10.45	+	+	-	+
15	(*E*)-4,8-Dimethyl-1,3,7-nonatriene	10.44	-	+	-	-
16	(*Z*)-3-Hexen-1-yl acetate	10.61				+
17	6-Methyl-5-hepten-2-one	11.01	+	+	+	-
18	(*Z*)-3-Hexen-1-ol	11.69	+	+	+	+
19	Nonanal	11.98	+	+	+	-
20	1,3,8-*p*-Menthatriene	12.68				-
21	*cis*-Linalool oxide	12.81	+	+	+	+
22	*trans*-Sabinene hydrate	13.08	+	+	+	+
23	Bicycloelemene	13.55	+	+	+	-
24	*α*-Copaene	13.78	+	+	+	+
25	*α*-Gurjunene	14.42	+	+	+	+
26	Fenchol	15.09	+	-	-	+
27	Calarene	15.49	+	+	+	+
28	*β*-Caryophyllene	15.57	-	+	+	+
29	Aromadendrene	15.79	+	+ *	+	+
30	*allo*-Aromadendrene	16.33	+ *	+ *	+ *	+
31	*trans*-Pinocarveol	16.34	+ *	+ *	+ *	+
32	*cis*-Verbenol	16.35	-	+ *	+ *	+
33	*δ*-Terpineol	16.52	+	+	-	-
34	*trans*-Verbenol	16.68	-	+	+	+
35	*α*-Terpineol	16.94	+ *	+ *	+ *	+
36	*α*-Terpinyl acetate	17.02	+ *	+ *	+ *	+
37	Ledene	17.11	+ *	+ *	+ *	+
38	2-Oxo-1,8-cineole	17.20	-	+	+	+
39	Verbenone	17.31	-	+	+	+
40	*exo*-2-Hydroxycineole acetate	17.42	+	-	-	-
41	2-*β*-Hydroxy-8-cineole	17.44	-	+	+	-
42	Bicyclogermacrene	17.66	+	+	+	-
43	*δ*-Cadinene	17.93	+	+	-	-
44	Ethyl 2-phenylethanoate	18.30	+	-	-	-
45	7-Hydroxy-1,8-cineole	18.36	-	+ *	+ *	-
46	Myrtenol	18.38	+	+ *	+ *	+
47	2-Phenethyl acetate	18.74	+	+	+	-
48	*trans*-Carveol	18.90	+	+	-	+
49	Geraniol	19.00	+	+	-	+
50	2-*α*-Hydroxy-1,8-cineole	19.31	-	+ *	+ *	+
51	9-Hydroxy-1,8-cineole	19.40	-	+ *	+ *	+
52	Benzyl alcohol	19.56	+	-	-	-
53	3-*α*-Hydroxy-1,8-cineole	19.96	-	+	+	-
54	2-Phenylethanol	20.05	+	+	-	+
55	Phenylethyl isovalerate	21.13	+	+	+	+
56	*epi*-Globulol	21.48	+	+	+	+
57	Globulol	22.32	+	+	-	+
58	Rosifoliol	22.65	+	+	-	-
59	Spathulenol	22.89	+	+	+	-
60	*tau*-Cadinol	23.42	+	+	-	-
61	*iso*-Spathulenol	24.01	+	+	-	-

Note: Total of 34 antennal recordings: 11 obtained with an SDE extract of *E. globulus*, 13 with a HS-Monotrap™ extract produced by *G. platensis* males in contact with *E. globulus* leaves and shoots and 10 obtained with a HS-Monotrap™ extract produced by *G. platensis* males.

## Data Availability

Not applicable.

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
