# Peer review of "Unveiling Chemical Cues of Insect-Tree and Insect-Insect Interactions for the Eucalyptus Weevil and Its Egg Parasitoid by Multidimensional Gas Chromatographic Methods"

_molecules, 2022, doi:10.3390/molecules27134042_

Round 1
Reviewer 1 Report
These are my main comments on the manuscript (molecules-1766131) entitled “Unveiling Chemical Cues of Insect-Tree and Insect-Insect Interactions for the Eucalyptus Weevil and its Egg Parasitoid by Multidimensional Gas Chromatographic Methods”. The manuscript investigates the compounds emitted by host plants of the eucalyptus weevil, Gonipterus platensis, might mediate host selection behaviour. Also, the pheromone blend of G. platensis is identified and active blend was achieved, composed by cis and trans-verbenol, verbenene, myrtenol and trans-pinocarveol. Following substantial revisions should be incorporated in the manuscript prior to acceptance.
1. I have concerns about the manuscript sections that I believe need to be addressed in order to improve its clarity.
2. Manuscript contains typos and not carefully and well prepared.
3. A hypothesis for this research is needed.
4. Results and discussion should be divided in two sections, without this the manuscript cannot published.
5. Other revisions could be checked in PDF attached.

Author Response
file uploaded

Reviewer 2 Report
In the paper Unveiling chemical cues of insect-tree and insect-insect interactions for the eucalyptus weevil and its egg parasitoid by multidimensional gas chromatographic methods, authors report a very good paper concerning a multidimensional gas chromatography as a powerful analytical tool.
Paper is veri interesting, good write and supported by results.
I suggest minor revision
Page 2 line 84
One-dimensional Gas Chromatography (1D-GC), is the most widely used technique for the analysis of volatiles and semi volatile compounds in complex biological matrices, such as the above mentioned plant bouquets or essential oils.
Please add refercences such as Flavouring extra-virgin olive oil with aromatic and medicinal plants essential oils stabilizes oleic acid composition during photo-oxidative stress. Agriculture, 2021, 11.3: 266.
Author Response
file uploaded

Reviewer 3 Report
See attach

Author Response
file uploaded
